# Unmasking the impact: Racial microaggressions and the health consequences for Latinas in the United States

Jeannine Ríos[1]*, Mindy Menn[2], George King[2], Trey L. DeJong[3], John Terrizzi[4], Ann Oyare Amuta-Jimenez[5]

**1** Department of Sociology and Public Health, The University of Texas at Dallas, Richardson, Texas, United States of America, **2** Department of Health Promotion and Kinesiology, Texas Woman's University, Denton, Texas, United States of America, **3** Department of Mathematics and Statistics, Washington State University, Pullman, Washington, United States of America, **4** Department of Psychology & Philosophy, Texas Woman's University, Denton, Texas, United States of America, **5** Department of Kinesiology and Public Health Division, The University of Texas, Arlington, Texas, United States of America

* Jeannine.Rios@UTDallas.edu

## Abstract

Researchers conducted an exploratory analysis of the relationship between perceived racial microaggressions and physical and mental health outcomes among Latinas living in the United States. The history of discrimination in the US has created lasting health disparities. Racial microaggressions, subtle forms of discrimination directed towards people of color, have become ubiquitous in the U.S. Research focused on Latinas is limited. This study used a descriptive and analytical cross-sectional design, collecting data online from December 2021 to February 2022, using the Racial Microaggressions Scale (RMAS). A Canonical Correlation Analysis (CCA) was conducted using the six RMAS sub-scales (Foreigner, Sexualization, Criminality, Low Achieving, Invisibility and Environmental) and three DASS (Depression, Anxiety and Stress). The sample included 659 self-identified Latinas' data representing 36 states. The full model was statistically significant *Wilks's λ* = .907 criterion, *F* (18, 1887.05) = 3.65, *p* < .001. with the $r^2$ type effect size was.093. Another CCA was conducted using the six microaggressions sub-scales previously mentioned as predictors of physical health outcomes using the Physical Health Questionnaire (PHQ) which has four sub-scales. This model was also statistically significant using the *Wilks's λ* = .872 criterion, *F*(24, 2286.23) = 3.82, *p* < .001. This study found that perceived racial microaggressions in the forms of Low Achieving and Sexualization were associated with negative health outcomes for Latinas in the forms of increased anxiety, and gastrointestinal problems. In addition, when Environmental microaggressions are present the overall effects are lessened. In addition, when Environmental microaggressions are present the overall effects are lessened. Findings revealed that perceived racial microaggressions, particularly Low Achieving and Sexualization, are associated with negative health outcomes for

**Data availability statement:** The data has been provided in a data repository OSF Storage https://osf.io/xyv76/?view_only=a08ad3a97e-5b4eb6aeb4c2c55abad3b6.

**Funding:** This work was supported by Texas Woman's University, Center for Student Research ($500 to JR). The funders had no role in the study design, data collection and analysis, decision to publish, or preparation of the manuscript.

**Competing interests:** The authors have declared that no competing interests exist.

Latinas, particularly including increased anxiety, and gastrointestinal issues, while Environmental microaggressions mitigated these effects.

## Introduction

Racial microaggressions are subtle forms of discrimination directed towards people of color [1–4]. Throughout the literature, interpersonal racial microaggressions are usually unconscious, making them difficult to preemptively recognize and defend [1–5]. Although quite damaging, blatant racial discrimination can be defended through legal action, whereas subtle microaggressions cannot; the invisibility of microaggressions multiplies their power [1–3,6]. One of the constant challenges with racial microaggressions is that they are missed even by the offender [3]. Sue (2010), stated that if microaggressions were obvious, it would be easier for the victims to acknowledge them and respond [3–5,7,8].

The study of racial microaggression research has been growing since Sue, et al. [9] created a taxonomy of racial microaggressions in everyday life in 2007. Racial microaggression research has typically concentrated on Black communities, creating a gap in studies involving Latino communities, particularly Latinas. The long history of discrimination in the US has resulted in damaging and lasting effects which influence inequities and health disparities. Qualitative research has been crucial in the study of marginalized communities, allowing for more in-depth analysis of individuals who are not represented in the literature. As microaggression research has continued to grow, a common perception among the research community has been to value quantitative research over qualitative research. Both are valid and necessary to gather information on the association between microaggressions and mental and physical health outcomes [4]. The current researchers chose a quantitative design to explore hypotheses with objective measurements to be applied to a larger sector of the Latina population.

Critical Race Theory (CRT), Latino Critical Race Theory (LatCrit) and Racial Battle Fatigue (RBF) theory have been used to understand the negative and racially charged experiences of People of Color in the United States. CRT is crucial in understanding and transforming power by studying race and racism and how power is maintained in society. CRT questions our systems and laws that maintain a structure of inequality based on race. Over the years marginalized groups of races other than Blacks have formed CRT specific to their ethnicities and causes to better address their concerns. For example, Latinos and Asian Americans created both Asian American jurisprudence and LatCrit to study immigration policy and language rights which CRT does not cover [10].

RBF is a theoretical framework whose foundation rests in the areas of sociology, social psychology, and racial stress in health [11,12]. Key components of this framework include everyday discriminations and racist exchanges that Latinos are subjected to [11–13]. The RBF model shows that microaggressions influence stress levels for individuals psychologically, physiologically, and behaviorally [3,8,11,13]. Victims of racial microaggressions often experience perpetual coping and defending

against microaggressions, which can lead to physical and emotional exhaustion [3,8,11,12]. Throughout the literature, RBF has been studied almost exclusively at college campuses with samples of undergraduate college students, where students of color were in a specific environment they could not control or abandon without exceptional consequences [11,12,14–16]. Due to differential power dynamics and social structures rampant in American higher education, college students must often deal with daily microaggressions which can lead to and exacerbate RBF [11–14].

The health effects associated with microaggressions are varied, cumulative, and well-documented [4,5,9]. Correlated health effects from microaggressions include emotional stress including depression, anxiety, psychotic symptoms, and physical stress including physical attacks, and somatic symptoms including headaches, indigestion, chest pains, hives, and fatigue. These stressors can lead to increased alcohol and smoking, cardiovascular problems, increased cortisol outputs, and obesity [7,17–24].

In work and academic situations where racial microaggressions occur by familiar perpetrators, Latinos and Blacks often feel vulnerable or hopeless in confronting aggressors [24,25]. In an environment where there are constant messages of disrespect and treatment of disregard through microaggressive acts, this is arguably a chronic stressor contributing to health disparities for Latinos [26].

Latinos are among the largest and fastest-growing ethnically marginalized groups in the United States [27,28]. The Latino population has been disproportionately affected by health disparities across multiple social determinants of health [29,30]. Social determinants of health are the conditions that one is born into and have a significant impact on one's health [31]. The most essential determinants for good health include adequate housing, safe neighborhoods, equitable jobs and wages, quality education, and equity in access to quality health care [27,28]. Latinas are the women subgroup represented among a group of Latinos. This sub-group within the Latino population encounters challenges faced by the intersectionality of two marginalized identities, being a woman and a Latino in the U.S. Intersectionality refers to both identities and systems of oppression that connect and impact structural inequalities. This research aimed to investigate the psychological dynamics of racism. An exploratory analysis of the relationship between perceived racial microaggressions and physical and mental health outcomes experienced by Latinas living in the United States. For this paper we explored two research questions 1. "Are microaggressions related to mental health status including depression, anxiety, and stress among Latinas?" and 2. " Are microaggressions related to physical health status among Latinas?"

## Methods

### Ethics statement

Institutional Review Board (IRB) approval was obtained from Texas Woman's University (TWU)-IRB, approval number IRB-FY2021–181, for all study-related materials. The study was a quantitative correlation study. All subjects provided their formal written informed consent for inclusion in the study before they participated. This included voluntary participation, anonymity, and the right to withdraw at any time.

### Selection and description of participants

We used original data obtained from self-identified Latinas using Prolific, TWU students, and three social media platforms (Linked In, Facebook/Meta, and Instagram). Prolific provided incentives to participants which was different than community and university sampling. Most of the participants were recruited by Prolific, a for-profit online participant recruitment company committed to improving research and finding representative samples based on study criteria [32]. Prolific paid each participant after completing this study's survey. Each participant was paid approximately $3.17 for 20 minutes at the rate of $9.50 per hour. For this study we specifically targeted Latinas in our messaging on social media platforms and via email messages. According to pretesting results, completing the survey took participants between a few minutes to 25 minutes. The average time to complete the survey was approximately 13 minutes.

## Data collection and measurement

Using the Racial Microaggression Scale (RMAS) [33], the Depression Anxiety Stress Scale (DASS) [34] and the Physical Health Questionnaire [PHQ; 35] data was collected online December 2021 to February 2022, 963 self-identified Latinas representing 18 countries and 36 states within the United States, participated in this study. Six hundred and fifty-nine participants' data were used for analyses from U.S. participants. One of the limitations of this study was that it only included participants from the US.

## Statistics

The RMAS 6 microaggression sub-scales were developed to assess themes and categories of microaggressions encountered by People of Color and some specific to Latinos: Foreigner/Not Belonging – assessed the theme of being treated as a foreigner in own land, or made to feel she/he is not a "true" American or is an outsider; Criminality- these assessed incidents in which one is treated as aggressive, dangerous or likely to engage in criminal behavior; Sexualization – assessed the theme of women of color being exoticized, being treated in an overly sexual manner due to one's race; Low-Achieving/ Undesirable Culture – this category assessed being treated as intellectually inferior and pathologizing cultural values and communications styles including being asked to play down one's culture; Invisibility -assessed being ignored at work or school because of one's race, expected to assimilate to White culture and being treated poorly due to race; and Environmental Invalidations – which assessed the perception of negative environmental messages derived from observing that powerful and visible roles in one's community are not held by individuals of one's own ethnicity or racial background or being the only person of color in certain settings [33]. The statistical software used was SPSS version 25.

The DASS 21-item questionnaire was designed to measure the severity of core symptoms of depression, anxiety, and stress. Each of the three DASS21 scales contains seven items per scale. The 7 -item Depression scale assesses hopelessness, devaluation of life, dysphoria, self-depreciation, lack of interest/involvement, anhedonia, and inertia. The Anxiety scale assesses autonomic arousal, skeletal muscle effects, situational anxiety, and subjective experience of anxious affect. The 7 -item Stress scale assesses difficulty relaxing, nervous arousal, and being easily agitated, irritable and impatient. Participants were asked to use a 4-point (1–4) severity/frequency scale to rate the amount they experienced each item in the week prior to survey participation. The items were summed to get a total score with higher scores indicating higher levels of psychological distress.

The current researcher used the PHQ 14-item scale to measure physical health status by examining participants' somatic symptoms. The four PHQ sub-scales tested were gastrointestinal problems, headaches, sleep disturbance, and respiratory infections. Potential scores can range from 12 to 98 on a 7-point Likert scale ranging from 1 (*not at all*) to 7 (*all the time*) with higher scores representing more somatic symptoms [36].

A priori power analysis was conducted using G*Power 3.1.9 to determine the minimum sample size required to find statistical significance using Pearson's correlation analysis. With a desired level of power set at.80, an alpha (α) level at.05, and a moderate effect size of.30 (ρ), a minimum of 84 participants was required to ensure adequate power. With a desired level of power set at.80, an alpha (α) level at.05, and a moderate effect size of.30 (*f*), a minimum of 111 participants were required to ensure adequate power [37]. Therefore, to ensure adequate power for all analyses and to account for dropouts, incomplete, or invalid cases, a minimum sample of 140 participants was planned for recruitment [37]. The online survey was completed by 788 participants. The survey included 15 main questions, 9 demographic questions, and three scales, which will be covered in this paper (RMAS, DASS and PHQ). The scales varied in length (e.g., RMAS- 32 items, DASS- 21 items, and PHQ- 14 items). 129 participants were removed who lived outside of the US which led to 659 participants' data included in the data analyses.

The RMAS has been tested for convergent validity, internal reliability, and concurrent validity among diverse samples. The validity and reliability test results demonstrated that the RMAS was a valid and reliable tool to assess the perceptions of racial microaggressions in a sample of 377 Black, Latino, Asian American, Asian, South Asian, Middle Eastern, and

multiracial-identifying adults between the ages of 18 and 76 years of age [33]. Cronbach's alpha was conducted on the 32-item scale among the diverse sample described with high internal consistency revealed for each sub-scale: Environmental Invalidations ($\alpha$ = .81); Foreigner/Not Belonging ($\alpha$ = .78); Low-Achieving/Undesirable Culture ($\alpha$ = .87); Criminality ($\alpha$ = .85); Sexualization ($\alpha$ = .83), and Invisibility ($\alpha$ = .89) [33]. Each scale in this study was tested for psychometric properties using Cronbach's Alpha (see Table 1).

A Canonical Correlation Analysis (CCA) was set up to analyze the latent variables among six sub-scales within the Racial Microaggression Scale (RMAS) used to measure microaggressions (Foreigner, Criminality, Sexualization, Low Achieving, Invisibility, and Environmental sub-scales) and three sub-scales within the Depression Anxiety Stress Scale-Short Form (DASS-21) used to measure mental health status [33,34]. CCA is a multivariate analysis of correlation used for these specific situations where multiple X and multiple Y correlations exist [38]. CCA determines a set of canonical variates, independent linear combinations of the variables within each set that best explain the variability both within and between groups [38,39]. In multiple regression analysis multiple independent variables are combined into a synthetic combination to best correlate with the dependent variable, with CCA this happens on both sides of the independent and dependent variables [38,40]. This allows for the strength of both sides of the relationship to be examined at the same time.

The independent variables for both research questions were the six racial microaggression sub-scales (Foreigner/ Not Belonging, Criminality, Sexualization, Low Achieving/Undesirable Culture, Invisibility and Environmental Invalidations) and the dependent variables were depression, anxiety, and stress. The independent variables, racial microaggression sub-scales, were measured with 32 items on a 4-point Likert scale for *how often* 1 (*never*) to 4 (*often/ frequently*) and for *how stressful, upsetting, or bothersome is this* (0 = *never happened to me*, 0 = *not at all*, 1 = *a little*, 2 = *moderate level*, 3 = *high level*). The scores from the 32 items were summed based on the categories of their sub-scales Foreigner (3 items), Criminality (4 items), Sexualization (3 items), Low Achieving (9 items), Invisibility (8 items), and Environmental (5 items) and the

**Table 1. Cronbach's Alphas for RMAS, DASS, and PHQ Sub-Scales.**

| Sub-Scales | Cases | Cronbach's Alpha | Number of Items |
|---|---|---|---|
| *RMASa** | 683 | .95 | 32 |
| Foreigner | 691 | .92 | 3 |
| Criminality | 691 | .86 | 4 |
| Sexualization | 689 | .89 | 3 |
| Low Achieving | 685 | .89 | 9 |
| Invisibility | 687 | .90 | 8 |
| Environmental | 688 | .79 | 5 |
| *DASS* | 787 | .94 | 21 |
| Depression | 787 | .92 | 7 |
| Anxiety | 787 | .78 | 7 |
| Stress | 787 | .86 | 7 |
| *PHQ* | 675 | .86 | 14 |
| Sleep Disturbances | 678 | .76 | 4 |
| Headaches | 678 | .87 | 3 |
| Gastrointestinal Problems | 677 | .85 | 4 |
| Respiratory Infections | 676 | .74 | 3 |

*Note.** RMASa = RMAS used in all RMAS analyses, it represents the frequency of racial microaggressions. RMASb = a measure of distress within the RMAS scale. RMASb was not used in these analyses.

means with standard deviations were calculated. The second part of this scale, *how stressful, upsetting or bothersome*, was not used to calculate further analyses.

The dependent variables for the second question were the physical health sub-scales as measured by using items from the adapted Physical Health Questionnaire (PHQ 14-item scale) [36]. The independent variables, racial microaggressions sub-scales, were measured in the same manner as research question 1. The dependent variable was measured on a 7-point Likert scale ranging from 1 (*not at all*) to 7 (*all of the time*). The scores from the 14 items were summed to create a physical health score for each participant. Potential scores ranged from 12 to 98, with lower scores representing fewer somatic symptoms and better physical health (see Table 2).

## Results

Most of the participants were between the ages of 18–24 years (55.7%) and 25–34 years (30.7%) reflective of the US population of Latinos which has a median age of 29.5 years in 2021 [41], female (95.3%), single or never married (79.1%) the number of unmarried Latinas has been declining over the years 47.3% in 2024 [42], self-identified Latina/o/x/His-panic (100%), heterosexual or straight (61.7%), bachelor's degree or higher (47.3%), employment status revealed that the majority of the participants were working either part or full time (49.1%) or students (34.4%) and unemployed (13.7%) with a household income at or below $30,000 (52.6%). Participants included two non-binary, one non-binary AFAB (i.e., assigned female at birth, one agender and one transgender male participant (see Table 3 below). Reported household income showed that 52.6% of participants earned 30K or less annually which is lower than national data of a median household income for Latinos of $59K annually [41]. Latinas, on their own continue to lag behind Latino men earning 85 cents for every dollar earned by Latino men and 62 cents compared to non-Hispanic White men [43]. Latinas are earning bachelor's degrees at a higher rate of 23% in 2023, than a decade prior in 2013 at 16% [43]. The majority of Latinos in the US speak English 72% of Latinos 5 years and older and 91% of US born Latinos [41]. Participants in this study were from 36 states in the US, with the majority living in Texas.

CCA was conducted using the six RMAS sub-scales as predictors of the DASS to evaluate the multivariate shared relationship between the two variable sets (i.e., racial microaggressions and mental health*). When conducting the

**Table 2. Canonical Solutions for RMAS Predicting Physical Health for Function 1.**

| Variable | Function 1 | | |
|---|---|---|---|
| | *Coef* | $r_s$ | $r_s^2$ (%) |
| **RMAS** Foreigner | -.121 | .435 | 18.92 |
| Criminality | .187 | .585 | 34.22 |
| Sexualization | .583 | .797 | 63.52 |
| Low Achieving | .686 | .785 | 61.62 |
| Environmental | -.499 | .206 | 4.24 |
| Invisibility | .074 | .561 | 31.47 |
| $R_c^2$=.097 | | | |
| **PHQ** Sleep Disorders | .278 | .631 | 39.81 |
| Headaches | .451 | .788 | 62.09 |
| Gastrointestinal | .474 | .815 | 66.42 |
| Respiratory | .158 | .525 | 27.56 |

*Note. Coef* = standardized canonical function coefficient; $r_s$ = structure coefficient; $r_s^2$ = squared structure coefficient.

**Table 3. Demographic Frequency Table.**

| Participant Characteristic | | |
|---|---|---|
| **Characteristic** | **Frequency** | **Percent (%)** |
| *Age* | | |
| 18-24 | 439 | 55.7 |
| 25-34 | 242 | 30.7 |
| 35-44 | 66 | 8.4 |
| 45-54 | 27 | 3.4 |
| 55-64 | 12 | 1.5 |
| 65-75 | 2 | 0.3 |
| *N* | 788 | 100 |
| *Sex* | | |
| Male | 29 | 3.7 |
| Female | 751 | 95.3 |
| Intersex | 1 | 0.1 |
| Other | 5 | 0.6 |
| Prefer not to answer | 2 | 1.5 |
| *N* | 788 | 100 |
| *Marital Status* | | |
| Never married | 167 | 21.2 |
| Married | 135 | 17.7 |
| Single | 456 | 57.9 |
| Divorced | 16 | 2 |
| Single, previously married | 13 | 1.6 |
| Missing n | 1 | 0.1 |
| *N* | 787 | 99.9 |
| *Ethnicity* | | |
| Ethnicity | | |
| Latinx, Latina/o, Hispanic | 788 | 100 |
| *Highest Education Level Completed* | | |
| Less than high school | 2 | 0.3 |
| High school graduate/GED | 105 | 13.3 |
| Some college | 126 | 16 |
| Associate degree | 60 | 7.6 |
| Bachelor's degree | 287 | 36.4 |
| Master's degree | 69 | 8.8 |
| Doctoral degree | 8 | 1 |
| Professional degree (JD, MD) | 9 | 1.1 |
| Trade school certificate | 6 | 0.8 |
| Trade school courses | 2 | 0.3 |
| Missing *n* | 114 | 14.5 |
| *N* | 674 | 100 |
| *Sexual Orientation* | | |
| Heterosexual/Straight | 486 | 61.7 |
| Transgender | 0 | 0.0 |
| Cisgender | 19 | 2.4 |
| Gender Fluid | 6 | 0.8 |
| Gay | 2 | 0.3 |

*(Continued)*

**Table 3.** (Continued)

| Characteristic | Frequency | Percent (%) |
|---|---|---|
| *Participant Characteristic* | | |
| **Characteristic** | **Frequency** | **Percent (%)** |
| Asexual | 14 | 1.8 |
| Bisexual | 126 | 16 |
| Pansexual | 16 | 2.0 |
| Lesbian | 21 | 2.7 |
| Queer | 12 | 1.5 |
| Questioning | 18 | 2.3 |
| Other | 7 | 0.9 |
| Prefer not to answer | 2 | 0.3 |
| Missing | 59 | |
| *N* | 729 | 92.5 |
| *Employment Status* | | |
| Employed Full time | 237 | 30.1 |
| Employed Part-time | 150 | 19 |
| Unemployed | 108 | 13.7 |
| Retired | 4 | 0.5 |
| Student | 271 | 34.4 |
| Missing n | 18 | |
| *N* | 770 | 97.7 |
| *Household Income* | | |
| Below $10,000 | 154 | 22.9 |
| *$10,001 to $20,000* | *118* | 17.6 |
| $20,001 to $30,000 | 81 | 12.1 |
| $30,001 to $40,000 | 39 | 5.8 |
| $40,001 to $50,000 | 19 | 2.8 |
| $50,001 to $60,000 | 76 | 11.3 |
| $60,001 to $75,000 | 50 | 7.4 |
| $75,001 to $100,000 | 54 | 8 |
| $100,001 to $150,000 | 38 | 5.7 |
| $150,001 or more | 43 | 6.4 |
| Missing *n* | 116 | |
| *N* | 672 | 97.7 |

analysis between the RMAS and DASS, 112 cases were rejected due to missing data leaving a sample size of 547 for this analysis.

The dimension reduction analysis showed that a statistically significant relationship existed between RMAS sub-scales and the DASS sub-scales. The analysis yielded three functions with squared canonical correlations ($R_c^2$) .075, .027, and .009 for each successive function. Collectively, the full model across all functions was statistically significant using the Wilks's $\lambda = .892$ criterion, $F(18, 1522.18) = 3.49$, $p < .001$. Because Wilks's $\lambda$ represents the variance unexplained by the model, $1 - \lambda$ yields the full model effect size in an $r^2$ metric. Thus, for the set of three canonical functions, the overall $r^2$ type effect size was .108, which indicates a medium effect size, the full model explained about 10.8% of the variance shared between the variable sets. Only the first canonical variate was interpreted because Function 1 was statistically significant with an $r^2$ of .075 indicating that it explained 7.5% of the variance shared between the variable sets. Function

2 was statistically significant but had a negligible effect size and was not interpreted, with Wilks's λ = .964 criterion, F(10, 1078.00) = 1.99, p = .032, and Function 3 was not significant Wilks's λ = .991 criterion, F(4, 540) = 1.26, p = .278. The $R^2_c$ .027 and.009 for the last two functions only explained 2.7%, and.9% of shared variance, respectively, which represented negligible effects.

For Function 1 Standardized canonical coefficient predictor variate, the strongest RMAS sub-scale scores were Sexualization (Coef = .36), Low Achieving (Coef = 1.0), and Environmental (Coef = -.48).. These were all primary contributors to the predictor variables. Upon further investigation the structure coefficients ($r_s$) for the predictor variables demonstrated the primary predictors were still Low Achieving ($r_s = .82$), and Sexualization ($r_s = .65$), and that on their own Criminality ($r_s = 57$) and Invisibility ($r_s = .42$) had a larger role, but environmental ($r_s = .16$) does not add much to the relationship on its own. Reviewing the functions shows that RMAS Sexualization and Low Achieving were the most relevant criterion, along with environmental inversely.

Regarding the Standardized canonical coefficients for criterion/outcome variable set in Function 1, Depression (Coef = .51), Anxiety (Coef = .59), and Stress (Coef = .00), Anxiety and Depression were the main outcome variables. DASS structure coefficient outcome variables were all high, Depression ($r_s = .89$), Anxiety ($r_s = .92$), and Stress ($r_s = .80$), but their standardized scores showed that together Depression and Anxiety are the relevant coefficients (See Table 4).

## Mental health

Overall, both Sexualization and Low Achieving had a positive relationship with DASS outcome variables, but Environmental had the opposite effect, refining the model by acting as a suppressor. Latinas reporting higher frequencies of racial microaggressions in these two forms (Sexualization and Low Achieving) reported higher levels of anxiety. However, when environmental microaggressions were present the effects of the entire model were lessened. As shown by investigating standardized and structure coefficients above, the relationship found here is primarily Sexualization, Low Achieving, and Environmental from RMAS and Depression and Anxiety from DASS. Overall, both Sexualization and Low Achieving had a positive relationship with predicted DASS outcome variables in the model, but environmental had the opposite effect, refining the relationships by acting as a suppressor. Latinas with higher frequency of racial microaggression in these two forms had higher levels of depression and anxiety. However, when environmental microaggressions were present the

**Table 4. Standardized and Structure Correlation Coefficients for RMAS Sub-Scales.**

| Variable | Function 1 | | |
|---|---|---|---|
| | Coef | $r_s$ | $r_s^2$ (%) |
| *RMAS* | | | |
| Foreigner | -.048 | .400 | 16.00 |
| Criminality | .320 | .575 | 33.06 |
| Sexualization | .359 | .652 | 42.51 |
| Low Achieving | 1.00 | .815 | 66.42 |
| Environmental | -.480 | .160 | 2.56 |
| Invisibility | -.342 | .417 | 17.38 |
| *DASS* | | | |
| Depression | .511 | .891 | 79.38 |
| Anxiety | .590 | .919 | 84.45 |
| Stress | .001 | .803 | 64.48 |

*Note.* Coef = standardized canonical function coefficient; $r_s$ = structure coefficient; $r_s^2$ = squared structure coefficient.

$R_c^2$ =.075 for the model

predicted levels of depression and anxiety decreased. Suppressors act to help the model fit better, even if they have no direct relationship with the outcome. Although Stress on its own could explain most of the model, Depression and Anxiety explained much of what Stress could explain as well as unique variance in the model. In other words, when environmental microaggressions are considered with Sexualization and Low Achieving microaggressions they more accurately predict DASS.

Essentially, if Latinas have a high frequency of microaggressions in the forms of Sexualization and Low Achieving and Environmental microaggressions are present they are predicted to have lower depression and anxiety compared to having high frequency of Sexualization and Low Achieving microaggressions without or with lower Environmental microaggressions, which would lead to more depression and anxiety according to the model.

## Physical health

Analyses that focused on microaggressions and physical health found that overall, both Low Achieving and Sexualization had a positive relationship with PHQ outcome variables, but environmental had the opposite effect refining the relationship acting as a suppressor again. Latinas with higher frequency of racial microaggression in these two forms reported higher levels of Gastrointestinal Problems. However, when environmental microaggressions were present the effects of the entire model were lessened.

A CCA was conducted using the six RMAS microaggression variables as predictors of the four physical health variables to evaluate the multivariate shared relationship between the two variable sets (i.e., microaggressions and physical health). There were 536 cases accepted, 123 cases were rejected due to missing data. The analysis yielded four functions with squared canonical correlations $R_c^2$ of .097, .029, .019, and .003 for each successive function. The full model across all functions was statistically significant using the *Wilks's λ* = .857 criterion, $F(24, 1836.20)$ = 3.46, $p < .001$. Because *Wilks's λ* represents the variance unexplained by the model, 1- λ yields the full model effect size in an $r^2$ metric. Thus, for the set of four canonical functions, the overall $r^2$ type effect size was .143, which indicates that the full model explained about 14.3% of the variance shared between the variable sets. The first two functions were statistically significant. Function 2 was significant, but the effect size was very small, Wilks's λ = .949 criterion, $F(15, 1455.22)$ = 1.85, $p = .024$; Dimension reduction analysis showed that Functions 3–4 were not significant Wilks's λ = .978 criterion, $F(8, 1056)$ = 1.49, $p = .155$; and Wilks's λ = .997 criterion, $F(3, 529)$ = 0.48, $p = .695$. The last three functions explained 2.9%, 1.9%, and .3% of shared variance.

The standardized canonical coefficients from RMAS sub-scales showed the major predictor variables to be Low Achieving (*Coef* = .69), Sexualization (*Coef* = .58), and inversely related Environmental (*Coef* = -.50). This conclusion was supported by the structure coefficients where Sexualization ($r_s$ = .80) and Low Achieving ($r_s$ = .79) were still the primary predictor variables (See Table 2). Although Environmental became less relevant, Criminalization, Invisibility, and Foreigner all became stronger predictor variables on their own. Much of what was explained by these three variables was shown to be explained by Low Achieving and Sexualization.

The major predictor variables for the standardized canonical coefficients of the PHQ were primarily Gastrointestinal Problems (*Coef* = .47) with secondary contributions from Headaches (*Coef* = .45). When investigating the structure coefficients, Gastrointestinal Problems is still the most relevant ($r_s$ = .82) with Headaches ($r_s$ = .79) adding more to the model on their own. In addition, Sleep Disorders ($r_s$ = .63) and Respiratory Infections ($r_s$ = .52), both contributed more to the model on their own (See Table 2).

As shown by investigating standardized and structure coefficients above, the relationship found here is primarily Low Achieving, Sexualization, and Environmental from RMAS, Gastrointestinal Problems and Headaches from PHQ. Overall, both Low Achieving and Sexualization had a positive relationship with PHQ outcome variables, but Environmental had the opposite effect refining the relationship acting as a suppressor. Latinas with higher frequency of racial microaggression in these two forms had higher levels of Gastrointestinal Problems and Headaches. However, when Environmental microaggressions were present the effects of the entire model were lessened. In other words, when environmental

microaggressions are present with Low Achieving and Sexualization microaggressions they more accurately predict PHQ. Essentially, if Latinas experience high microaggressions in the forms of Sexualization and Low Achieving and Environmental microaggressions are present they are predicted to have lower Gastrointestinal Problems and Headaches compared to having high Sexualization and Low Achieving microaggressions with fewer Environmental microaggressions, which would predict more Gastrointestinal Problems and Headaches according to the model. It could be that Environmental microaggressions act as a protective factor for Gastrointestinal Problems and Headaches.

## Discussion

The two research questions were 1. "Are microaggressions related to mental health status including depression, anxiety, and stress among Latinas?" and 2. " Are microaggressions related to physical health status among Latinas?" The primary researcher studied the relationship between racial microaggressions and mental and physical health outcomes among Latinas (See Table 4). The results showed that racial microaggressions had a statistically significant impact on mental health for this population in the form of increased anxiety. When examining racial microaggressions and physical health outcomes the results showed a statistically significant increase in gastrointestinal problems among Latinas experiencing this type of discrimination. The primary researcher also found that when in the presence of environmental microaggressions, the effects of other microaggressions and negative health outcomes decreased.

Environmental microaggressions also called environmental invalidations are forms of discrimination that add to an individual's sense of invisibility. Present in situations where there is no one else from your race or ethnicity in positions of power, when one is treated as if they are not noticed, or they are dismissed [33]. Our findings suggest that environmental microaggressions could lessen the impacts of racial microaggressions in instances of Low achieving and Sexualization microaggressions among Latinas Table 5.

This could suggest that when Latinas are aware of Environmental microaggressions the effects of other microaggressions, and their negative health outcomes decreases. For example, if a Latina knows in advance, that she will be entering a meeting where there are all White men, it can be less anxiety producing than if she did not have those environmental cues at all. She can mentally prepare herself for a potential negative outcome. Although these specific outcomes have not been mirrored within the literature, there is research to support increased group identification when faced with perceived discrimination among racial minority groups, also known as the Rejection-Identification model (RIM) [34,44,45]. The RIM indicates that perceived discrimination may lead to increased group identification, which can support psychological well-being in the presence of societal devaluation [34,44,45]. In their research, they found that multiracial identification is a type of coping response used by People of Color to reduce the overall cost of discrimination on well-being [46]. Environmental microaggressions may be acting as protective factors by reminding Latinas of their group identity. They may also be acting as an alert for other racial microaggressions. Emotional regulation is part of the RIM as individuals re-appraise their environment based on social cues exposed by environmental microaggressions. For example, as a Latina enters a board room and notices the environmental microaggression of being the only female, woman of color, she may begin to reinterpret emotions and understand that she will need to suppress certain aspects of who she is as she begins to code switch [47,48].

**Table 5. Scales Used to Measure Mental and Physical Health Related to RMAS.**

| Scale | Health Outcomes Measured | # Of Items |
|---|---|---|
| DASS (3 sub-scales: Depression, Anxiety, Stress) | Mental Health | 21 |
| PHQ (4 sub-scales: Sleep, Headaches, Gastrointestinal, Respiratory) | Physical Health | 14 |
| RMAS (6 sub-scales: Foreigner, Criminality, Sexualization, Low Achieving, Environmental and Invisibility) | Racial Microaggressions | 32 |

Research by Nadal (2018) and Manohar (2015) suggests that effective coping mechanisms encompass social support from family, friends, and affinity groups, as well as developing an awareness of racial microaggressions. Such sources of support and understanding can enhance one's comprehension of the challenges tied to systemic oppression and discrimination [49,50]. In addition, Manohar (2015) found that emotional support and social identity affirmation support in coping with everyday racial discrimination could be a helpful coping strategy.

Moreover, across message types, participants perceived high-quality messages as being more effective generally, more effective in achieving multiple goals, and in facilitating emotional improvement and collective esteem enhancement. Also, for person-centered messages, reappraisal and empathy mediated the influence of message quality on message effectiveness, multiple goal achievement, and affect change. While in the case of social identity affirmation messages, reattribution mediated the influence of message quality on message evaluation. Finally, source ethnicity and participants' reported number of inter-ethnic friends emerged as significant moderators [50].

Due to the nature of this study, researchers were able to utilize CCA, which may have facilitated the discovery of the role of environmental microaggressions as a suppressor that might not have been identified otherwise. More studies are needed to clarify the mechanism of this relationship. Additionally, Spanish-speaking Latinas approached the primary researcher to express their interest in participating in this study. Although the study was not initially designed for Spanish speakers, there is a clear need and desire to include monolingual Spanish-speaking Latinas in future research efforts.

Online surveys can increase access for many and be cost efficient way to collect data [51]. However, online surveys can be a limitation by introducing bias for those participants self-selecting themselves into the sample, who are literate in English and have access to the internet. Although the current researchers contracted with Prolific to gather most of the participants who had been vetted by the company, participants self-selected into the study. Future similar studies are needed in Spanish to capture the monolingual Spanish speakers among Latinas and the Latino community.

## Public health implications

The potential implications of these findings can be used for racism prevention initiatives. In addition, adding accurate narratives about Latinos and other racially marginalized groups can help these individuals and groups be better understood and valued throughout our society. Initiatives such as education and policies that require a deeper understanding of all people in our society could increase knowledge and support among racially marginalized communities. For example, ethnic studies courses and a comprehensive study of histories of all cultures in academia could improve behaviors. In the workplace, education about implicit biases, microaggressions and systems of oppression can raise awareness and with accountability measures correct these behaviors. This could decrease biases and systems of oppression. As marginalized people are valued, individuals may be less likely to devalue their own culture in order to be accepted by mainstream white centered values [3,4,8].

Implications of this exploratory analysis are promising for the field of microaggression research and especially for the Latina population and other marginalized communities in the US. Marginalized communities in the US tend to represent the global majority which has even larger implications when studying microaggressions worldwide [52]. While certain microaggressions are specifically and uniquely rooted in American culture, the concepts of differential treatment, disproportional power dynamics, race-based conflicts, and racial microaggressions are unfortunately pervasive across cultures, nations, and continents. The need to educate communities about microaggressions, especially those who hold power in society, can lead to a shift in culture and a decrease in microaggressive acts, ultimately leading to better mental and physical health outcomes [9].

## Acknowledgments

The main author would like to thank her mother, Nellie Rios, and her daughter Ayanna Reyes for their unconditional love and support.

## Author contributions

**Conceptualization:** Jeannine Rios.

**Data curation:** Jeannine Rios.

**Formal analysis:** Jeannine Rios, Trey L. DeJong.

**Funding acquisition:** Jeannine Rios.

**Investigation:** Jeannine Rios.

**Methodology:** Jeannine Rios.

**Supervision:** Mindy Menn, George King, Trey L. DeJong.

**Validation:** Trey L. DeJong.

**Writing – original draft:** Jeannine Rios.

**Writing – review & editing:** Jeannine Rios, Mindy Menn, George King, John Terrizzi Jr, Ann Oyare Amuta-Jimenez.

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
