## [Decision Letter · Decision Letter 0]

12 Aug 2024

PMEN-D-24-00063

Unmasking the Impact: Racial Microaggressions and the Health Consequences for Latinas in the United States

PLOS Mental Health

Dear Dr. Rios,

Thank you for submitting your manuscript to PLOS Mental Health. After careful consideration, we feel that it has merit but does not fully meet PLOS Mental Health’s publication criteria as it currently stands. Therefore, we invite you to submit a revised version of the manuscript that addresses the points raised during the review process.

We look forward to receiving your revised manuscript.

Kind regards,

Sugy Choi

Academic Editor

PLOS Mental Health

Additional Editor Comments (if provided):

Reviewers' comments:

Reviewer's Responses to Questions

**Comments to the Author**

1. Does this manuscript meet PLOS Mental Health’s publication criteria ? Is the manuscript technically sound, and do the data support the conclusions? The manuscript must describe methodologically and ethically rigorous research with conclusions that are appropriately drawn based on the data presented.

Reviewer #1: Partly

Reviewer #2: Yes

Reviewer #3: Partly

2. Has the statistical analysis been performed appropriately and rigorously?

Reviewer #1: Yes

Reviewer #2: Yes

Reviewer #3: Yes

3. Have the authors made all data underlying the findings in their manuscript fully available (please refer to the Data Availability Statement at the start of the manuscript PDF file)?

Reviewer #1: No

Reviewer #2: No

Reviewer #3: Yes

4. Is the manuscript presented in an intelligible fashion and written in standard English?

Reviewer #1: Yes

Reviewer #2: Yes

Reviewer #3: Yes

5. Review Comments to the Author

Reviewer #1: I would like to extend my appreciation for the effort you have invested in this work.

- Expand the keywords.

- Ensure proper in-text citation formatting for all references.

- Clearly state the research aims, mentioning investigating the psychological dynamics of racism.

- In the introduction section, explain in detail about the research topic with reference to qualitative studies.

- Explain the method of determining the sample size.

- While you've mentioned the sample size and demographic characteristics, consider providing more details relevant to the research question.

- Further clarification is needed to explain the sampling methods used.

- What statistical solutions have you used to control intervening variables and increase internal validity?

- Is your sample representative of the entire population?

- Did you include population with a history of mental health issues?

- With what rationale have you chosen your sample from the countries and states mentioned?

- Each section can start with a brief introduction summarizing the analysis performed and then present the key findings.

- Explain about the limitations of online sampling in the discussion section.

- The explanation of the results can be richer.

- It seems that you can use interdisciplinary perspectives (cultural sociology and cultural - psychiatry) to explain the results obtained.

- Sample loss is one of the important limitations of your study. What statistical methods have you - used to prevent reducing the internal and external validity of the study?

- You can mention the psychometric properties (e.g., Cronbach's Alpha) in a separate paragraph or table.

- Briefly mention the content of the informed consent form, highlighting aspects like the voluntary nature, anonymity, confidentiality, and right to withdraw.

- In discussion briefly restate the research question and the theoretical framework used in the study.

- delve deeper into the limitations of theories in differentiating between races.

- Emphasize the potential implications of the findings for racism prevention strategies.

Reviewer #2: Thank you for the opportunity to peer-review “Unmasking the Impact: Racial Microaggressions and the Health Consequences for Latinas in the United States”. Herein, the authors describe the harm of microaggressions on racialized peoples in the U.S. Latinas within the U.S. completed three online surveys exploring microaggressions, mental well-being, and physical symptoms. A statistical analysis was completed which describes symptoms of anxiety and gastrointestinal issues as prevalent symptoms related to experiencing microaggressions. Significantly, the team identified that all symptoms were analyzed to be lessened in the context alongside environmental microaggressions.

This is a well-written piece, adding valuable evidence to a very timely and important discussion. The last paragraph on page 13 describes an example of what an environmental microaggression might constitute. It would be beneficial to consider adding a brief description or summary of what each of the 6 microaggression sub-scales entail early in the manuscript. Similar as to how the authors briefly described the SDoH on page 3. It will add context and help the manuscript stand on its own, as implications include educating communities on microaggressions.

Reviewer #3: Dear Authors, I congratulate you on coming up with a nice research topic & nice piece of work. Please find my comments below:

Methods: The aim & objectives are to be explicitly provided here (it's there in the abstract, though)

Results: There are some repetitions in the result section (repetition that sexualisation and low achievement have a positive influence on the model, while environmental factor has a negative influence on the overall model's effect size), which can be avoided

Discussion: The most unique finding of the paper is that the environment had the opposite effect, refining the relationship by acting as a suppressor. These findings need to be elaborated to highlight the context and perspective. I think talking about its items will also be useful for the readers. Regards, reviewer

6. PLOS authors have the option to publish the peer review history of their article (what does this mean? ). If published, this will include your full peer review and any attached files.

**Do you want your identity to be public for this peer review?** For information about this choice, including consent withdrawal, please see our Privacy Policy .

Reviewer #1: No

Reviewer #2: No

Reviewer #3: **Yes: ** DR Snehil Gupta

---

## [Decision Letter · Decision Letter 1]

7 Jan 2025

PMEN-D-24-00063R1

Unmasking the Impact: Racial Microaggressions and the Health Consequences for Latinas in the United States

PLOS Mental Health

Dear Dr. Rios,

Thank you for submitting your manuscript to PLOS Mental Health. After careful consideration, we feel that it has merit but does not fully meet PLOS Mental Health’s publication criteria as it currently stands. Therefore, we invite you to submit a revised version of the manuscript that addresses the points raised during the review process.

We look forward to receiving your revised manuscript.

Kind regards,

Sugy Choi

Academic Editor

PLOS Mental Health

Journal Requirements:

Additional Editor Comments (if provided):

Reviewers' comments:

Reviewer's Responses to Questions

**Comments to the Author**

1. If the authors have adequately addressed your comments raised in a previous round of review and you feel that this manuscript is now acceptable for publication, you may indicate that here to bypass the “Comments to the Author” section, enter your conflict of interest statement in the “Confidential to Editor” section, and submit your "Accept" recommendation.

Reviewer #4: (No Response)

Reviewer #5: (No Response)

2. Does this manuscript meet PLOS Mental Health’s publication criteria ? Is the manuscript technically sound, and do the data support the conclusions? The manuscript must describe methodologically and ethically rigorous research with conclusions that are appropriately drawn based on the data presented.

Reviewer #4: Yes

Reviewer #5: Yes

3. Has the statistical analysis been performed appropriately and rigorously?

Reviewer #4: Yes

Reviewer #5: Yes

4. Have the authors made all data underlying the findings in their manuscript fully available (please refer to the Data Availability Statement at the start of the manuscript PDF file)?

Reviewer #4: Yes

Reviewer #5: Yes

5. Is the manuscript presented in an intelligible fashion and written in standard English?

Reviewer #4: Yes

Reviewer #5: Yes

6. Review Comments to the Author

Reviewer #4: Thank you for your paper, Unmasking the Impact: Racial Microaggressions and the Health Consequences for Latinas in the United States. The paper is well written and describes research on a timely and important topic of microaggression’s association with health in an understudied population (Latinas). The revised paper is responsive to the reviewer’s comments. I suggest several minor revisions to improve clarity and flow.

Abstract

• Move the background info in the Methods (1st three sentences) to the Objective section as this presents the Background or Significance for your study, then your current study Objective sentence will flow nicely.

• Focus on the US sample only (36 states/659 participants) and moving this detail from Methods to the Results.

• Include DASS and PHQ in Methods (spell out instruments + acronyms) as a measure of mental health and physical health respectively, then sharing the findings as you currently have in the Results

• Move your analytic method CCA to last sentence in Methods (currently in Results)

• Move the specific findings on the RMAS subscales (1st two sentences) from Discussion to Results these are findings.

• Suggest adding a concluding sentence on implications of your findings for research or practice/policy – summarizing sentence from what you have in the main text

Introduction

• Suggest having your new section (The study of racial aggression has been growing since…” ) as a new paragraph

• In the Abstract you state that much of the research on microaggressions and health have been with Black populations and to a lesser extent Latinos and even lesser extend Latinas. Suggest describing this in your Intro to improve the transition between the paragraphs about what microaggressions are and their impacts on health and health disparities, and the paragraph on Latinos and Latinas.

• Suggest adding your study objective to the end of the Intro section – moving the research questions from the Methods section lines 170-175 to end the Introduction

Methods

• Include your study design

• Suggest including more info about sampling from your response to reviewers, namely about the purposive sampling method that was used and that you included community, university and prolific samples (how are prolific different than community and university)?

• Put the acronyms for the scales when you first introduce them so you can use the acronyms going forward

• Line 126 add “RMAS” to 6 microaggression subscales. Thank you for providing the detail about these, suggest listing the subscales for DASS and PHQ as well

• Please move Table 3 to the Methods section. You could also provide scoring in this table instead of text if space is an issue

• Suggest moving the scale scoring to earlier in the methods section where you describe each instrument, as well as whether higher scores denote better/worse health.

• Suggest focusing only on the scales and sample you used for this study, no need to reference the full sample and other scales as they do not pertain to your research questions.

• How long did it take to do the survey and did participants receive any incentives for participation?

• What statistical software did you use for analysis?

Results

• This paper focuses on Latinas so would expect that 100% of the sample is female or that you included Latinas who do not identify as female (eg gender non-binary) in the study? Please clarify

• Add in a new Table 1 with participant characteristics so the reader can get a fuller understanding of who participated – eg. You state that the sample was mainly students (34%) but we don’t know what other categories were included

• 18-34 is a broad category, please report any more specific age data if you have this available

• Describe how representative your sample is compared to English speaking Latinas in the US overall – you do this for income but not for age, marital status, education, etc

• Suggest adding sub-headers for mental health and for physical health and moving the last

two paragraphs of Results lines 328-338 to start off each section then go into detailed analysis

Discussion

• It’s not clear why LatCRT and RBF are starting out the Discussion section as theoretical frameworks typically are in the Introduction section to provide background/rationale for your study or Methods to provide justification for what variables/data you collected. As it reads now, these theories belong in Intro as RBF shows how racism can impact health. If you are including LatCRT and RBF to help explain or contextualize your findings, suggest editing the Discussion to link these theories to your findings

• Suggest adding the need to engage monolingual Spanish speaking Latinas in future research given your focus on microaggressions. Also given environmental microaggressions had a surprising finding, further research with Latinas from other environments (outside student status/universities) would be important to see whether this relationship persists

• The new text on 399 – 409 “The potential implications…” belongs in the later Public Health Implications section. Thank you for adding these practice implications

Reviewer #5: Two research questions were raised in this manuscript that sought to (1) determine the extent to which a relationship exist between racial microaggressions and mental health status and (2) establish whether or not racial microaggressions were related to physical health among 659 Latinas participants. The independent variables for both research questions were six racial microaggression sub-scales (Foreigner/ Not Belonging, Criminality, Sexualization, Low Achieving/Undesirable Culture, Invisibility, and Environmental Invalidations) and the dependent variables were depression, anxiety, and stress that were measured by the Depression, Anxiety, and Stress Scale (DASS). The dependent variables for the second research question were the physical health status with four sub-scales (Gastrointestinal Problems, Headaches, Sleep Disorders, and Respiratory Infections) measured by the Physical Health Questionnaire (PHQ).

A Canonical Correlation Analysis (CCA) was set up to analyze the latent variables among six sub-scales within the Racial Microaggression Scale (RMAS) used to measure microaggressions (Foreigner, Criminality, Sexualization, Low Achieving, Invisibility, and Environmental sub-scales), and three sub-scales within the Depression Anxiety Stress Scale-

(DASS) was used to measure mental health status. CCA determines a set of canonical variates, independent linear combinations of the variables within each set that best explain the variability both within and between groups.

Findings from the study indicated that a statistically significant relationship was found to exist between the RMAS sub-scales and the DASS sub-scales; and the six RMAS microaggression variables as predictors of the four physical health variables used to evaluate the multivariate were also found to have a relationship between the two variable sets (i.e., microaggressions and physical health). The full model across all functions was statistically significant. It is noted that the standardized canonical coefficients from RMAS sub-scales showed the major predictor variables to be Low Achieving (Coef =.69), Sexualization (Coef =.58), and inversely related to Environmental (Coef =-.50). This conclusion is supported by the structure coefficients where Sexualization ( =.80) and Low Achieving ( =.79) were still the primary predictor variables as in Table 2. Although Environmental became less relevant, Criminalization, Invisibility, and Foreigner all became stronger predictor variables on their own. Much of what was explained by these three variables was shown to be explained by Low Achieving and Sexualization. The major predictor variables for the standardized canonical coefficients of the PHQ were primarily Gastrointestinal Problems (Coef = .47) with secondary contributions from Headaches (Coef = .45). When investigating the structure coefficients, Gastrointestinal Problem is still the most relevant ( = .82) with Headaches ( = .79) adding more to the model on their own. In addition, sleep disorders ( = .63) and Respiratory Infections ( = .52), both contributed more to the model on their own.

As demonstrated by the standardized and structure coefficients, the relationship found is primarily Low Achieving, Sexualization, and Environmental from RMAS, Gastrointestinal Problems, and Headaches from PHQ. Overall, both low-achieving and Sexualization had a positive relationship with PHQ outcome variables, but Environmental had the opposite effect refining the relationship by acting as a suppressor. It is observed from the results that Latinas with a higher frequency of racial microaggression in these two forms were found to have higher levels of Gastrointestinal Problems and Headaches. However, when Environmental microaggressions are present the effects of the entire model are reduced, the results of this study are therefore found to be supportive of the conclusions.

Racial microaggressions have a damaging effect on members of marginalized and ethnic minority groups, and qualitative research strategy is the popular tool for unearthing psychological and emotional reactions that occur in response to the experience of microaggression, including anger, sadness, frustration, stigma, discrimination, and isolation. And for this study to venture into this phenomenon by quantitative method is commendable. Although the results of this study may not be generalizable to other populations given that it is a cross-sectional study, it certainly has implications for marginalized populations and ethnic minority groups globally.

Notwithstanding the merits of this manuscript, the following concerns have been noted for the attention of the authors:

1. This study reflects a descriptive and analytical cross-sectional design, and it is suggested that as part of the abstract and in particular the methodology section the design of the study should be stated to guide readers.

2. It is noted further that as part of the findings of the study, the environmental sub-scale as part of the independent variables (RMAS), is found to act as a suppressor and protector on both the DASS and Physical Health Status thereby decreasing the overall effect of DASS and PHQ as evidenced in the results of the study. For instance, Latinas with a higher frequency of racial microaggression in these two forms had higher levels of depression and anxiety. However, when environmental microaggressions were present the predicted levels of depression and anxiety decreased. Suppressors act to help the model fit better, even if they have no direct relationship with the outcome. This finding is supported by theory, and it is suggested that this theory should be cited especially under the discussion section to back the claim.

Environments are constantly changing, and as such, human beings have ample opportunity to adapt to different situations and self-regulate their behavior, cognitions, and affect in a way that is consistent with their goals. The area of emotion regulation is subsumed under the broader domain of self-regulation. Several reasons may account for these results. First, people who tend to reappraise situations may be more likely to appraise or reappraise their social interactions as positive and enjoyable, leading to more positive responses on measures of social well-being. In addition, reappraisal use is positively correlated with positive affect which in turn is associated with higher interpersonal functioning and greater social satisfaction. Lastly, when negative social interactions arise, the use of reappraisal may help reduce negative emotional reactivity, allowing for a more civil and less hostile interaction, in turn reducing the likelihood of negative social experiences. Taken together, the research suggests that reappraisal may be an adaptive and helpful strategy to use in social situations (Ríos-Rodríguez et al., 2024). This may have implications for rolling out interventions on coping strategies among these clinical populations.

7. PLOS authors have the option to publish the peer review history of their article (what does this mean? ). If published, this will include your full peer review and any attached files.

**Do you want your identity to be public for this peer review?** For information about this choice, including consent withdrawal, please see our Privacy Policy .

Reviewer #4: **Yes: ** Lesley Steinman

Reviewer #5: No

---

## [Decision Letter · Decision Letter 2]

29 Jun 2025

PMEN-D-24-00063R2

Unmasking the Impact: Racial Microaggressions and the Health Consequences for Latinas in the United States

PLOS Mental Health

Dear Dr. Rios,

Thank you for submitting your manuscript to PLOS Mental Health. After careful consideration, we feel that it has merit but does not fully meet PLOS Mental Health’s publication criteria as it currently stands. Therefore, we invite you to submit a revised version of the manuscript that addresses the points raised during the review process.

Your revised manuscript has been closely examined by both external reviewers and our internal editors and although we are satisfied with most of your revisions, we require a greater emphasis on the responses and edits in accordance with Reviewer 5's comments.

Specifically, we have identified the following comments made by Reviewer 5 which have not been fully addressed by the authors in their revision. Particularly, the reviewer requests a fuller discussion of what might be considered a more counter-intuitive finding, and suggests a theoretical basis for it. The relevant section of their comments is reproduced below:

"Notwithstanding the merits of this manuscript, the following concerns have been noted for the attention of the authors: 1. This study reflects a descriptive and analytical cross-sectional design, and it is suggested that as part of the abstract and in particular the methodology section the design of the study should be stated to guide readers.

"2. It is noted further that as part of the findings of the study, the environmental sub-scale as part of the independent variables (RMAS), is found to act as a suppressor and protector on both the DASS and Physical Health Status thereby decreasing the overall effect of DASS and PHQ as evidenced in the results of the study. For instance, Latinas with a higher frequency of racial microaggression in these two forms had higher levels of depression and anxiety. However, when environmental microaggressions were present the predicted levels of depression and anxiety decreased. Suppressors act to help the model fit better, even if they have no direct relationship with the outcome. This finding is supported by theory, and it is suggested that this theory should be cited especially under the discussion section to back the claim.

"Environments are constantly changing, and as such, human beings have ample opportunity to adapt to different situations and self-regulate their behavior, cognitions, and affect in a way that is consistent with their goals. The area of emotion regulation is subsumed under the broader domain of self-regulation. Several reasons may account for these results. First, people who tend to reappraise situations may be more likely to appraise or reappraise their social interactions as positive and enjoyable, leading to more positive responses on measures of social well-being. In addition, reappraisal use is positively correlated with positive affect which in turn is associated with higher interpersonal functioning and greater social satisfaction. Lastly, when negative social interactions arise, the use of reappraisal may help reduce negative emotional reactivity, allowing for a more civil and less hostile interaction, in turn reducing the likelihood of negative social experiences. Taken together, the research suggests that reappraisal may be an adaptive and helpful strategy to use in social situations (Ríos-Rodríguez et al., 2024). This may have implications for rolling out interventions on coping strategies among these clinical populations."

We look forward to receiving your revised manuscript.

Kind regards,

Jenna Scaramanga

Staff Editor

PLOS Mental Health

Journal Requirements:

Reviewers' comments:

Reviewer's Responses to Questions

**Comments to the Author**

1. If the authors have adequately addressed your comments raised in a previous round of review and you feel that this manuscript is now acceptable for publication, you may indicate that here to bypass the “Comments to the Author” section, enter your conflict of interest statement in the “Confidential to Editor” section, and submit your "Accept" recommendation.

Reviewer #4: All comments have been addressed

2. Does this manuscript meet PLOS Mental Health’s publication criteria ? Is the manuscript technically sound, and do the data support the conclusions? The manuscript must describe methodologically and ethically rigorous research with conclusions that are appropriately drawn based on the data presented.

Reviewer #4: Yes

3. Has the statistical analysis been performed appropriately and rigorously?

Reviewer #4: Yes

4. Have the authors made all data underlying the findings in their manuscript fully available (please refer to the Data Availability Statement at the start of the manuscript PDF file)?

Reviewer #4: Yes

5. Is the manuscript presented in an intelligible fashion and written in standard English?

Reviewer #4: Yes

6. Review Comments to the Author

Reviewer #4: (No Response)

7. PLOS authors have the option to publish the peer review history of their article (what does this mean? ). If published, this will include your full peer review and any attached files.

**Do you want your identity to be public for this peer review?** For information about this choice, including consent withdrawal, please see our Privacy Policy .

Reviewer #4: **Yes: ** Lesley Steinman

---

## [Editor Report · Decision Letter 3]

6 Aug 2025

Unmasking the Impact: Racial Microaggressions and the Health Consequences for Latinas in the United States

PMEN-D-24-00063R3

Dear PH.D. Rios,

We are pleased to inform you that your manuscript 'Unmasking the Impact: Racial Microaggressions and the Health Consequences for Latinas in the United States' has been provisionally accepted for publication in PLOS Mental Health.

Best regards,

Karli Montague-Cardoso

Staff Editor

PLOS Mental Health